# Evaluating the Necessity of Adaptive RT and the Role of Deformable Image Registration in Lung Cancer with Different Pathologic Classifications

**DOI:** 10.3390/diagnostics13182956

**Published:** 2023-09-15

**Authors:** Woo Chul Kim, Yong Kyun Won, Sang Mi Lee, Nam Hun Heo, Seung-Gu Yeo, Ah Ram Chang, Sun Hyun Bae, Jae Sik Kim, Ik Dong Yoo, Sun-pyo Hong, Chul Kee Min, In Young Jo, Eun Seog Kim

**Affiliations:** 1Department of Radiation Oncology, Division of Medical Physics, Soonchunhyang University Cheonan Hospital, 31, Suncheonhyang 6-gil, Dongnam-gu, Cheonan 31151, Republic of Korea; veura@schmc.ac.kr (W.C.K.); arnold@schmc.ac.kr (C.K.M.); 2Department of Radiation Oncology, Soonchunhyang University Cheonan Hospital, 31, Suncheonhyang 6-gil, Dongnam-gu, Cheonan 31151, Republic of Korea; yong.won@schmc.ac.kr; 3Department of Nuclear Medicine, Soonchunhyang University Cheonan Hospital, 31, Suncheonhyang 6-gil, Dongnam-gu, Cheonan 31151, Republic of Korea; c91300@schmc.ac.kr (S.M.L.); 92132@schmc.ac.kr (I.D.Y.); 81907@schmc.ac.kr (S.-p.H.); 4Clinical Trial Center, Soonchunhyang University Cheonan Hospital, 31, Suncheonhyang 6-gil, Dongnam-gu, Cheonan 31151, Republic of Korea; hello3933@schmc.ac.kr; 5Department of Radiation Oncology, Soonchunhyang University Bucheon Hospital, 170, Jomaru-ro, Bucheon 14584, Republic of Korea; md6630@schmc.ac.kr (S.-G.Y.); baesh@schmc.ac.kr (S.H.B.); 6Department of Radiation Oncology, Soonchunhyang University Seoul Hospital, 59, Daesagwan-ro, Yongsan-gu, Seoul 04401, Republic of Korea; changaram@schmc.ac.kr (A.R.C.); icarusky@schmc.ac.kr (J.S.K.)

**Keywords:** lung cancer, radiotherapy, adaptive radiotherapy, deformable image registration, pathologic difference, radiation response

## Abstract

Background: This study aimed to analyze differential radiotherapy (RT) responses according to the pathological type of lung cancer to see the possibility of applying adaptive radiotherapy (ART). Methods: ART planning with resampled-computed tomography was conducted for a total of 30 patients (20 non-small-cell lung cancer patients and 10 small-cell lung cancer patients) using a deformable image registration technique to reveal gross tumor volume (GTV) changes according to the duration of RT. Results: The small-cell lung cancer group demonstrated an average GTV reduction of 20.95% after the first week of initial treatment (*p* = 0.001), whereas the adenocarcinoma and squamous cell carcinoma groups showed an average volume reduction of 20.47% (*p* = 0.015) and 12.68% in the second week. The application of ART according to the timing of GTV reduction has been shown to affect changes in radiation dose irradiated to normal tissues. This suggests that ART applications may have to be different depending on pathological differences in lung cancer. Conclusion: Through these results, the present study proposes the possibility of personalized treatment options for individual patients by individualizing ART based on specific radiation responses by pathologic types of lung cancer.

## 1. Introduction

Lung cancer, a common cancer worldwide, presents significant treatment challenges owing to its inherent pathological characteristics and complex clinical prognosis. According to the World Health Organization, lung cancer accounts for approximately 11.6% of all new cancer diagnoses, and it is one of the cancers with the highest mortality rate [1,2]. Lung cancer is pathologically divided into non-small-cell lung cancer (NSCLC) and small-cell lung cancer (SCLC), each with intrinsically different genetic and immunological characteristics [3,4]. NSCLC accounts for approximately 80–85% of all lung cancers, with adenocarcinoma and squamous cell carcinoma (SCC) being the most common [5,6,7]. These subtypes of NSCLC show differences in genetic mutations, phenotypes, and immune responses, which influence patient survival rates and treatment responsiveness [8]. In particular, adenocarcinoma often carries specific genetic mutations such as those in EGFR, ALK, and ROS1, which can be targeted therapy, offering an effective treatment strategy in combination with radiotherapy (RT) [9]. SCLC constitutes about 15–20% of lung cancers and is characterized by a high proliferation rate and malignancy of small cells [10]. These characteristics lead to an increased response to radiation in SCLC; however, rapid recurrence and a high propensity for metastasis contribute to a relatively low overall survival rate, necessitating more intensive treatment.

For lung cancer, RT is effectively combined with chemotherapy; however, the RT approach does not vary significantly according to the pathological subtype [11]. Because NSCLC and SCLC differ considerably in their pathological characteristics and treatment responses, personalized RT plans are necessary. Evaluating and understanding adaptive therapy for lung cancer based on the characteristics of each subgroup is vital [12]. However, our understanding of how these pathological differences connect to responsiveness of adaptive therapy remains limited. During RT, geometric changes in the treatment target due to patient respiration occur frequently and significantly influence the effectiveness of the treatment. As RT progresses, the size, shape, and position of the tumor change over time, complicating optimization of the treatment plan and potentially hindering predictable RT outcomes [13,14].

This study aims to investigate the differences in RT responses according to the pathological classification of lung cancer, and the geometric changes that occur during RT that influence these responses. Based on this understanding, we aimed to evaluate the necessity and possibility of adaptive RT (ART) for different lung cancer subtypes to more accurately track changes during RT and apply them to treatment plans. By evaluating the role of deformable image registration (DIR), we aimed to develop a method for accurately capturing these geometric changes and quickly reflecting them in treatment plans. Ultimately, this study aimed to develop and implement a personalized RT strategy that is best suited for individual patients with lung cancer, thereby improving their survival rates and quality of life.

## 2. Materials and Methods

### 2.1. Patient Selection and Grouping

This study investigated 418 patients diagnosed with lung cancer who underwent RT at our institution between January 2016 and December 2019. This study was approved by the Institutional Review Board of Soonchunhyang University Cheonan Hospital (protocol number: SCHCA 2023-07-055) and was conducted according to the guidelines of the Declaration of Helsinki. The requirement for patient consent was waived by the Institutional Review Board because of the retrospective nature of the study.

The following were the patient selection criteria: patients (i) who received RT for radical purposes, with a prescribed daily dose of 2 Gy and reduced field (RF) plan of at least 20 times; (ii) who underwent cone beam computed tomography (CBCT) every time they received treatment; (iii) whose planning target volume (PTV) was included in the CBCT images including at least five image slices both above and below; and (iv) whose treatment was completed without interruption. The exclusion criteria were as follows: patients (i) with a history of RT to the lung or mediastinum due to other tumors and (ii) whose homogeneity index (HI) and conformity index (CI) in the resampled-computed tomography (r-CT) plan differed by more than 2% from the initial plan.

Finally, a total of 30/418 patients were enrolled based on the aforementioned criteria, and the patients were divided into three subgroups according to lung cancer type: adenocarcinoma, SCC, and SCLC subgroups (n = 10).

### 2.2. Patient Simulation and Treatment Planning Process

Generally, patients who received treatment >20 times, as counted from the initial treatment plan, were identified and were observed for at least 3 weeks before implementing the RF plan. The details of the equipment used are provided in Appendix A.

### 2.3. Imaging Acquisition and Registration Process

Initial CT images for simulation and CBCT images were collected from three groups of patients. Initial CT was used to establish the initial treatment plan, while CBCT images, including the image taken on the first day of RT, were collected at one-week intervals, four per patient. The acquired CBCT and initial CT images were transferred to MIM software Version 6.8.10 (MIMSoftware, Beachwood, OH, USA) for DIR. DIR was performed on CBCT and initial CT using MIM software. DIR was performed using VoxAlign Algorithms, which demonstrate locational accuracy within an average of 1.2 mm in the lung [15]. During this process, the deformed initial CT image resampled based on the CBCT-was extracted from the corresponding resampled CT (rCT) images for each CBCT (Figure 1).

### 2.4. Treatment Plan Application and Recalculation

The rCTs extracted from the MIM software were transferred to a RT planning system (Eclipse, version 16.1, Varian Medical System, Palo Alto, CA, USA). All organs on the rCT, excluding the GTV, were contoured using Manteia software (AccuContour, version 1.1, Manteia, Milwaukee, Brookfield, WI, USA) to maintain consistency using standardized methods. For lesions in upper and middle lobes with less breathing movement, the PTVs were summed to include all internal target volumes (ITVs), and the plan was executed accordingly. For lesions in the lower lobe, considering respiratory movements, the treatment plan was established by summing ITVs at 30–70 phases. In all rCT images, the GTV and PTV were modified for each image by a single experienced radiation oncology specialist (Figure 2). The treatment plan was established with intensity-modulated RT using seven beams of the optimal angles with energy of 6 MV. Optimization and calculation were performed to align each patient’s initial treatment plan and the treatment plan for rCT as closely as possible, thereby executing each ART plan (e.g., rCT1 for the ART1 plan). The prescription dose for the PTV was set identically to each patient’s initial plan, with a total of 40–44 Gy in 20–22 fractions, adjusted to include 95% of the PTV.

### 2.5. Evaluation of Treatment Plan Consistency

To validate the adaptive RT plans, a rigorous verification step was performed to ascertain the consistency between the initial treatment plan established from the initial CT and the treatment plans from the total of four rCTs obtained at one-week intervals. This crucial step ensures the reliability of each treatment plan and enhances the accuracy of subsequent analyses. The verification process was based on key dosimetric indicators: the conformality index (CI), radical dose homogeneity index (rDHI), and moderate dose homogeneity index (mDHI) [16]. The calculation methods for each index are provided in Appendix A.

### 2.6. Data Collection and Parameter Setting

We analyzed four instances of data obtained through each patient’s treatment plan and r-CT; these are listed in Appendix A. Changes in other critical organs at risk (OAR) for each rCT were analyzed using a dose–volume histogram (DVH); absolute organ volumes receiving more than 20 Gy (V20, cm^3^), absolute organ volumes receiving more than 30 Gy (V30, cm^3^), and mean dose (Dmean, Gy) of the total lung; the maximum and mean dose (Dmax and Dmean, Gy) of the esophagus; and the V20, V30, absolute organ volumes receiving more than 40 Gy (V40, cm^3^), and mean dose (Dmean, Gy) of the heart.

### 2.7. Statistical Analysis

We used SPSS 27.0, a statistical software package, to analyze the patient characteristics. Categorical variables were analyzed using Pearson’s chi-square test or Fisher’s exact test. Continuous variables were analyzed using Student’s *t*-test if they satisfied the normal distribution through a normality test, and the Mann–Whitney U test was used if they did not. The results for categorical variables are expressed as n (%). Continuous variables are expressed as mean ± standard deviation (SD) for Student’s *t*-test, and as median (interquartile range (IQR: from the first quartile to the third quartile)) for the Mann–Whitney U test. For statistical analysis of treatment plan, GTV, and OAR evaluations, we performed a repeated-measures ANOVA to assess the differences over time. The results are expressed as mean ± standard deviation (SD), and statistical significance was set at *p* ≤ 0.05.

## 3. Results

### 3.1. Patient Characteristics

This study included 30 patients: NSCLC, n = 20; and SCLC, n = 10. The average age of the patients was 75 (54–89) years, with 25 males (83.33%) and five females (16.67%). The interval between capturing the initial CT and the start of the actual RT was an average of 3.5 (2–6) days (Table 1).

### 3.2. Evaluation of Treatment Plan Consistency

We compared the treatment plans of all groups using the mean CI, mDHI, and rDHI index to verify the consistency between each patient’s initial plan and ART plans based on rCT. The CI value was 1.05 ± 0.00 on average in all groups, showing high statistical homogeneity with a difference from the initial plan of −0.01% ± 0.06 (initial plan vs. ART1, *p* > 0.999; initial plan vs. ART2, *p* > 0.999; initial plan vs. ART3, *p* > 0.999; initial plan vs. ART4, *p* > 0.999). The average mDHI value was 0.96 ± 0.01, showing high statistical homogeneity with a difference from the initial plan of −0.22% ± 0.53 (initial plan vs. ART1, *p* = 0.655; initial plan vs. ART2, *p* = 0.123; initial plan vs. ART 3, *p* > 0.999; and initial plan vs. ART4, *p* = 0.554). The rDHI value was on average 0.90 ± 0.03, showing high statistical homogeneity with a difference from the initial plan of −0.35% ± 0.64 (initial plan vs. ART1, *p* > 0.999; initial plan vs. ART2, *p* > 0.999; initial plan vs. ART3, *p* = 0.320; and initial plan vs. ART4, *p* = 0.686). The treatment plan was redesigned when a difference of >2% occurred in the CI, mDHI, and rDHI values from the initial plan in the patients’ ART plans. Consequently, the average CI, mDHI, and rDHI values were maintained at appropriate levels of 1.052 ± 0.001, 0.963 ± 0.011, and 0.899 ± 0.027, respectively, in all groups, and the differences from the initial plan were on average −0.012% ± 0.061 (*p* > 0.05), −0.224% ± 0.527 (*p* > 0.05), and −0.354% ± 0.638 (*p* > 0.05), respectively (Table 2). Thus, there was no significant difference between the initial and ART plans. The overall similarity of the indices ensures consistency of the treatment plan and proves the reliability of the ART treatment plans.

Thus, the PTV parameters (CI, mDHI, and rDHI) of the initial plan and ART plans showed minimal differences of less than 2% on average in all groups, confirming excellent consistency in comparing the changes in surrounding OAR due to tumor changes (Figure 3).

### 3.3. GTV Changes

We analyzed the volume changes of the GTV over four weeks after completion of RT for three different types of lung cancer: adenocarcinoma, SCC, and SCLC. Tumor volume progression during the treatment period was analyzed from two perspectives.

#### 3.3.1. Weekly Stepwise Tracking of GTV Changes: A Cumulative Comparison from the Start of Treatment

Across all groups, the average GTV for all patients per week was measured as 56.68 ± 66.65 cm^3^ at rCT1, with subsequent decreases to 52.27 ± 68.95 cm^3^, 45.20 ± 63.88 cm^3^, and 42.21 ± 58.47 cm^3^ at rCT2, 3, and 4, respectively. Compared with rCT1, average volume change rates were significantly different: −7.77% at rCT2 (*p* = 0.010), −20.26% at rCT3 (*p* < 0.01), and −25.54% at rCT4 (*p* < 0.01) (Figure 4).

In the subgroup analysis, the weekly changes in GTV after the start of RT were compared. First, in the adenocarcinoma group, average volume changes rates compared to rCT1 were −5.86% at rCT2 (*p* > 0.999), −20.47% at rCT3 (*p* = 0.015), and −26.86% at rCT4 (*p* = 0.009). In the SCC group, average volume change rates were not significantly different compared with rCT1: −1.07% at rCT2, −12.68% at rCT3, and −19.03% at rCT4. In the SCLC group, average volume reduction compared to rCT1 was the most prominent among the three subgroups: −20.95% at rCT2 (*p* = 0.01), −33.51% at rCT3 (*p* < 0.01), and −36.14% at rCT4 (*p* > 0.01). A significant reduction was observed from the point of one-week post-treatment compared to the other groups (Table 3). These results confirmed that the GTV gradually decreased as radiation treatment progressed, particularly in the SCLC group.

#### 3.3.2. Weekly Interval Comparison of GTV Alterations

Second, we observed the alterations in GTV throughout the course of RT and investigated when major changes occurred according to each subgroup. Across all patients, an average reduction of approximately 7.77% was observed from rCT1 to rCT2, followed by an additional decrease of approximately 13.54% from rCT2 to rCT3. Finally, a further decrease of approximately 6.62% occurred from rCT3 to rCT4. This declining trend was statistically significant at all intervals, except for rCT3 to rCT4 (*p* < 0.001) (Figure 5).

In subgroup analysis, the adenocarcinoma group showed a significant overall decrease (*p* < 0.001) in GTV. Specifically, reductions of approximately 5.86%, 15.52%, and 8.03% were observed in rCT1 to rCT2 (not significant), rCT2 to rCT3 (*p* = 0.031), and rCT3 to rCT4 (not significant) intervals, respectively. Despite the consistent decrease, statistical significance was not observed. In the SCLC group, reductions of approximately 20.95%, 15.89%, and 3.95% were observed in the intervals rCT1 to rCT2, rCT2 to rCT3, and rCT3 to rCT4, respectively, and the overall decrease in GTV was statistically significant (*p* < 0.001) (Table 3). These results indicate that the volumetric changes in GTV in response to treatment vary by group, and, within each group, changes vary with time. In summary, we confirmed that the GTV changes occur on a weekly basis in all patients. Notably, adenocarcinoma and SCC exhibited the most significant changes in the rCT2 to rCT3 interval, whereas SCLC showed the most significant changes in the rCT1 to rCT2 interval. This indicates that the timing of response to radiation differs depending on the cancer type.

### 3.4. OAR

#### 3.4.1. Whole Lung

First, we analyzed the V20, V30, and D_mean_ for the lungs in all groups between the starting point of RT, ART1, and ART2-4. The average reductions from ART1 to ART2-4 of V20 were −4.30%, −9.31%, and −11.40% respectively. The average reductions from ART1 to ART2-4 of V30 were −6.11%, −10.56%, and −13.74% respectively. The average reductions from ART1 to ART2-4 of D_mean_ were −5.82%, −9.85%, and −10.42% respectively. All three parameter values (V20, V30, and D_mean_) from ART1 to ART4 in all groups decreased significantly as the treatment progressed (Table 4).

In a subgroup analysis for the adenocarcinoma group, average reduction in V20 from ART1 to ART2, 3, and 4 was −2.98%, −9.06%, and −9.57%, respectively. The average reductions in V30 from ART1 to ART2, 3, and 4 were −1.83%, −5.03%, and −10.26%, respectively.

In the case of the SCC group, average reductions in V20 from ART1 to ART2, 3, and 4 were −3.29%, −7.05%, and −12.64%, respectively. The average reduction in V30 from ART1 to ART2, 3, and 4 was −4.46%, −10.17%, and −13.94%, respectively. The average reductions in D_mean_ from ART1 to ART2, 3, and 4 were −6.09%, −13.13%, and −14.03%, respectively.

For the SCLC group, average reductions in V20 from ART1 to ART2, 3, and 4 were −6.35%, −11.71%, and −11.73%, respectively. The average reductions in V30 from ART1 to ART2, 3, and 4 were 11.48%, −15.75%, and −16.54%, respectively. The average reductions in D_mean_ from ART1 to ART2, 3, and 4 are −9.77%, −9.12%, and −9.66%, respectively.

In the SCLC group, all changes in V20, V30, D_mean_ from ART1 to ART4 were statistically significant. However, despite the consistent decrease in ART1–4 levels in the adenocarcinoma and SCC groups, not all parameters showed significant results (Table 4).

#### 3.4.2. Esophagus

In the analysis of the D_max_ and D_mean_ values for all groups in the esophagus, there were variations from the ART1 plan to the ART4 plan, without significant differences. The D_max_ for all groups began at 29.16 ± 15.51 Gy in the ART1 plan and slightly decreased to 28.59 ± 15.18 Gy in the ART4 plan, but this was not significant (*p* = 0.933). Similarly, D_mean_ decreased from 12.57 ± 11.63 Gy in the ART1 plan to 10.78 ± 9.50 Gy in the ART4 plan, without significant difference (*p* = 0.731) (Table 5).

There were no statistically significant differences in D_max_ and D_mean_ of the esophagus from the ART1 plan to the ART4 plan in both the entire group and individual group analyses (Table 5).

#### 3.4.3. Heart

Heart analysis of all groups showed a statistically significant difference in V20 between ART1 and ART4. However, no statistically significant differences were found for V30 and D_mean_ when comparing between ART1 and ART4. In both the adenocarcinoma and SCC groups, there were no statistically significant changes in heart V20, V30, and D_mean_ values when comparing ART1 to ART4 (Table 6).

However, statistically significant differences were observed between V20 and V30 in the SCLC group. The results indicate that the radiation dose to the heart decreased over time with the ART plan in the SCLC group (Table 6).

## 4. Discussion

This study aimed to investigate the differences in RT responses according to the pathological classification of lung cancer, and to accordingly evaluate the necessity and possibility of ART for different lung cancer subtypes. We found that one week after the initiation of RT, the GTV decreased by 5.86%, 1.07%, and 20.95% in the adenocarcinoma, SCC, and SCLC groups, respectively, compared to that with ART1 (Table 2). This volume reduction was the most substantial in the SCLC group (*p* = 0.001). Our results confirm the findings of the previous studies, which show that SCLC possesses higher radiosensitivity compared to NSCLC.

Research concerning anticancer drugs for SCLC is significantly less common than that for NSCLC [11,17,18]. Treatment progression for SCLC has remained largely unchanged over the past 20 years, generally comprising 4–6 cycles of chemotherapy with cisplatin and etoposide, RT, and prophylactic cranial irradiation. Despite a consistent course of treatment, the median survival duration has improved to about 25–30 months, with approximately one-third of patients surviving beyond 5 years [19,20]. One of the reasons for this improvement in survival duration, despite the relatively unaltered treatment regimen for SCLC, is change in RT techniques and doses [19].

Historically, the radiation dose for SCLC has been 40–50 Gy either once a day (QD) or twice a day (BID). However, the INT-0096 clinical trial showed that most patients may relapse and suggested the need for an additional 60 Gy QD [19,21,22,23,24]. Conventionally, in both NSCLC and SCLC, the treatment regimen of 2 Gy QD typically leads to the application of a RF RT plan of approximately 40–44 Gy in 20–22 fractions. The objective of this approach is to mitigate potential side effects, such as radiation pneumonitis and esophagitis, by minimizing radiation exposure to surrounding normal tissues, such as the lungs and esophagus, as the tumor volume decreases. However, according to the changes in GTV observed in this study, proceeding with the initial treatment plan without modifications until the 20–22 fractions were administered could deprive the patient of opportunity for further reduction in radiation exposure to the surrounding normal tissue.

SCLC has a relatively faster cell cycle and higher growth rate compared to NSCLC. This implies a higher likelihood of more cells being positioned at the M and G2 stages of the cell cycle, where RT is most effective [25,26,27]. Furthermore, SCLC tends to have a richer blood supply than NSCLC, implying that oxygenated cells might impair DNA damage repair capacity, causing SCLC to be more responsive to radiation. These cellular and biological differences in SCLC have been similarly reported by several studies, suggesting that SCLC is more sensitive to radiation than NSCLC [12]. Bearing this in mind, the feasibility of treating patients with SCLC using the same RT techniques as that for NSCLC is questionable.

In the second week after the initiation of treatment, different volumetric reduction characteristics were observed in each group. A significant volumetric decrease of 20.47% was observed (*p* = 0.015) in the adenocarcinoma group, as opposed to approximately 12.68% in the SCC group, which was not significant. The relatively small volumetric reduction in the SCC group needs further investigation. The SCLC group continued to show a higher sensitivity than NSCLC, confirming a substantial volumetric decrease of 33.51% (*p* < 0.001). By the third week, the group-wise volumetric reductions were approximately 26.86% (*p* = 0.009) in adenocarcinoma, 19.03% (*p* = 0.758) in SCC, and 36.14% (*p* < 0.001) in SCLC. These variations illustrate the differences in treatment responses depending on the pathologic type of lung cancer, and different ART application times were identified according to the subgroups of lung cancer. With reduction in the GTV as RT progresses, radiation exposure of the surrounding normal tissues can be reduced to optimize treatment effects.

To validate the results of this study, it was necessary to demonstrate the similarity between the different treatment plans. For a tumor that changes temporally, it is essential to plan the ART such that distribution of radiation dose to the tumor matches the initial plan. CI, mDHI, and rDHI parameters were used to evaluate the similarity of the tumors in different treatment plans [16]. CI evaluates how well the distribution of radiation applied to the tumor site matches the intended plan, confirming whether the planned treatment has been applied correctly to the tumor site [28,29]. According to the RTOG guidelines, the closer the CI to 1, the more suitable the treatment plan for the tumor. Accurate investigation of the tumor site can minimize potential treatment side effects. HI is an indicator used to assess the consistency of a RT plan; it represents the consistency and uniformity of RT at the tumor site and evaluates the uniformity of the distribution of radiation within the tumor site. The closer the HI to 1, the more uniform the distribution of RT at the tumor site, which can promote effective control of the tumor site [28,30]. As presented earlier, each factor was planned to have a difference of <2%. This was based on the reliability of the dose distribution to other organs in the ART treatment plan.

Generally, the possibility of RT side effects can be evaluated in an initial treatment plan. However, the distribution of radiation dose changes due to changes in tumor volume and location and the movement of surrounding organs over the treatment period. Radiation pneumonitis is the most common side effect of RT in patients with lung cancer and is highly correlated with the volume of the irradiated lung [31,32,33]. While the decrease in GTV due to RT was not statistically significant in all patients, the GTV tended to decrease with the progression of the treatment period; accordingly, the radiation dose to the lung also tended to decrease on average. However, this correlation was not consistent in all groups; specifically, in the adenocarcinoma group, there was no clear correlation between GTV reduction and lung radiation dose reduction. In the adenocarcinoma group, the average GTV reductions in ART2, 3, and 4 compared to ART1 were −5.86%, −20.47%, and −26.86% respectively, but the decrease in radiation dose to the lung was not significant at V20 > 0.999, 0.943, and 0.051, respectively; at V30 > 0.999, 0.125, and 0.041, respectively; and at D_mean_ > 0.999, 0.619, and 0.710, respectively. In contrast, in the SCLC group, there was a close correlation between GTV reduction and lung radiation dose reduction. In the SCLC group, the average GTV reductions in ART2, 3, and 4 compared to ART1 were −20.95%, −33.51%, and −36.14% respectively. Concurrently, the lung radiation dose also decreased, showing statistical results at V20 of 0.481, 0.028, and 0.119; and at V30 of 0.007, 0.010, and 0.032, respectively. Notably, the SCLC group showed the largest decrease in GTV, by 20.96% in the first week compared to other groups, and the corresponding lung V20 value also decreased most significantly, from 364.07 ± 188.39 cm^3^ to 321.38 ± 153.20 cm^3^ (a decrease rate of −6.35%, *p* = 0.007). This confirms that the SCLC group responded the most to RT even after a week. According to Palma’s study, for the lung, the risk of radiation pneumonitis increases by 1.03 times for every 1% increase in V20 [32]. Applying these figures without considering chemotherapy or other factors, reducing V20 could potentially decrease the risk of radiation pneumonitis in patients with SCLC by approximately 22%. In addition, around the two-week point, the NSCLC groups began to exhibit a significant reduction in GTV. Correspondingly, this led to a decline in the V20, V30, and D_mean_ values of the lung. In contrast, when considering the average indices for the esophagus and heart in the adenocarcinoma and SCC groups, there were no significant differences. However, in the SCLC group, there was a trend towards a relative decrease in D_mean_ for the esophagus and in the V20 and V30 values for the heart in the ART3 and ART4 plans. This suggests that for the SCLC group, which had the largest GTV reduction, ART could potentially alleviate radiation-related side effects on the esophagus and heart to a certain degree.

Based on the evidence obtained in this study, we suggest individualized timing for the initiation of ART based on the pathological type of lung cancer. Considering the high sensitivity of SCLC to radiation, it may be effective to consider the application of ART as early as one week after the start of treatment. Conversely, in the adenocarcinoma group, although there was a comparable reduction in volume to that of SCLC in the second week, and the SCC group showed a smaller volume decrease relative to the other groups, the decrease was still observable. Therefore, for these two groups, it would be advisable to consider implementing ART before the beginning of the third week of treatment. However, adaptation of ART in RT for lung cancer has its limitations. Not all hospitals possess the equipment to apply ART to all treatments, and scheduling a new planning CT for every treatment plan without equipment could increase both the patient’s radiation exposure and the workload of the medical staff. DIR technology has been emphasized as a solution to these problems [34]. DIR considers the anatomical changes that occur with each treatment, and can quickly and efficiently apply these changes to existing treatment plans, enabling personalized treatment [35]. In particular, by applying DIR using CBCT or MVCT images captured during each treatment, ART can be performed without exposing the patient to additional radiation [36]. Therefore, DIR should be emphasized as an essential technology for achieving optimal treatment effects. Thus, we can improve the precision and stability of RT while minimizing additional radiation exposure to the patient and the workload of the medical staff. Ultimately, this will play a decisive role in providing the most appropriate RT.

This study had certain limitations. It was a retrospective study with a small number of participants, and many patients dropped out during the process of enrolling. This had limitations in terms of statistical robustness and limits the generalizability of the findings. Important characteristics such as T stage, chemotherapy, and GTV were heterogeneous among patient groups, potentially resulting in bias in the results, and future studies are required to control for these characteristics. Finally, concerns regarding adverse outcomes of lung cancer RT that may be caused by the early application of ART after reduction in GTV should be addressed. The process of expanding from GTV to clinical target volume (CTV) is designed to effectively administer treatment, including treatment for microscopic lesions. Of course, the application of ART due to changes of GTV owing to RT can dramatically reduce the side effects in normal organs; however, there are no studies on whether control of radiation in the subclinical region, which has escaped from the RT field through the application of early ART, is sufficient. Therefore, it is important to balance the risk of lesion recurrence with GTV reduction when applying different treatment approaches for SCLC and NSCLC. Accordingly, to prove the non-inferior treatment outcomes and safety of early ART compared to conventional RT techniques, more prospective studies, including systematic evaluation and management, are required.

## 5. Conclusions

This study systematically evaluated the utility of ART using DIR for lung cancer subtypes. We meticulously verified the sensitivity to radiation according to subtypes of lung cancer, while maintaining a consistent treatment plan. Furthermore, we observed changes in GTV due to treatment and its subsequent sparing effects on adjacent normal tissues. These findings underscore the importance of individualized treatment strategies and the potential value of ART. In particular, by developing personalized treatment strategies that consider sensitivity to radiation according to the pathological type of lung cancer, we explored the possibility of a more effective lung cancer treatment. Through future research, these methodologies are expected to contribute to the improvement of prognosis and maximization of treatment effects in patients with lung cancer. Accordingly, this study is anticipated to serve as a stepping stone to highlight the potential of ART in cancer treatment.

## Figures and Tables

**Figure 1 diagnostics-13-02956-f001:**
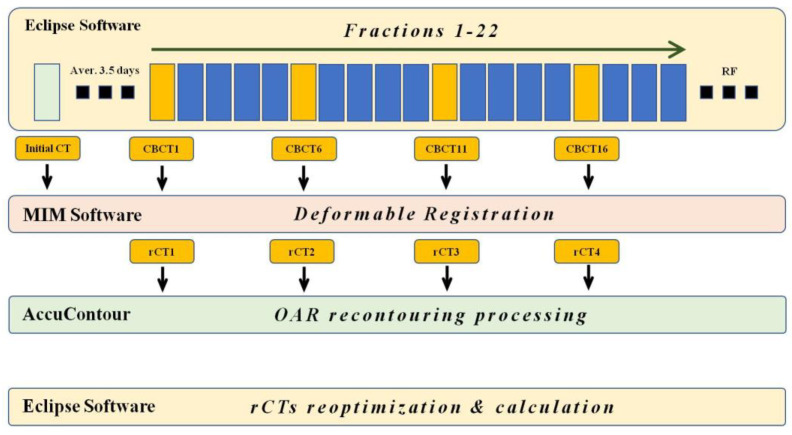
Schematic diagram of the imaging acquisition, registration, and RT planning workflow. The flowchart provides a simplified representation of the image acquisition and registration processes. PlanCT and CBCT were initially obtained using the Eclipse software, after which they were exported to the MIM software. Deformable registration and resampling were performed within the MIM. Then, using the Manteia software, the process of OAR recontouring is executed. Subsequently, the data were sent back to Eclipse for the final stages of re-optimization and calculation. RT, radiotherapy; RF, reduction field; OAR, organ at risk.

**Figure 2 diagnostics-13-02956-f002:**
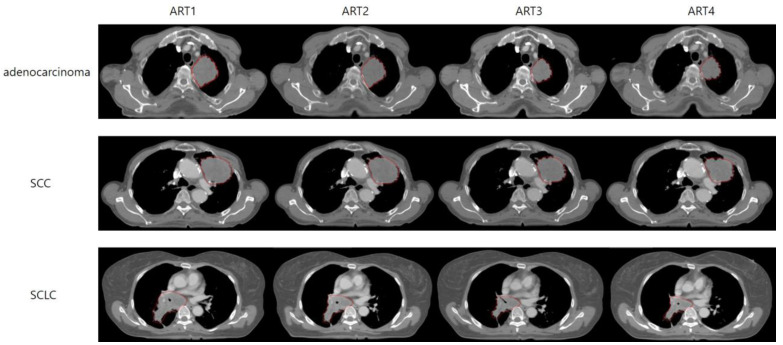
Demonstration of GTV reduction over ART stages in patients with adenocarcinoma, SCC, and SCLC. The figure displays a selection of CT images from one patient each for the adenocarcinoma, SCC, and SCLC groups. In each set of images, the gross tumor volume (GTV) at the same anatomical section is presented from the initial plan to ART4. SCC, squamous cell carcinoma; SCLC, small-cell lung cancer; ART, adaptive radiotherapy.

**Figure 3 diagnostics-13-02956-f003:**
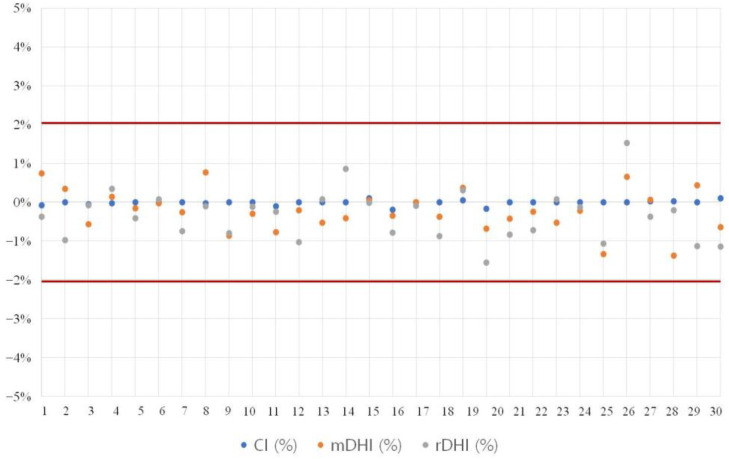
Comparison of conformity between initial plan and ART plans (CI, mDHI, rDHI). This figure illustrates the distribution of conformity index (CI), modified dose homogeneity index (mDHI), and relative dose homogeneity index (rDHI) for individual patients across the initial plan and the ART1, 2, 3, and 4 plans. The graph shows that the variance in these indices did not exceed 2% between the initial and subsequent ART plans, indicating high conformity across the treatment process. ART, adaptive radiotherapy.

**Figure 4 diagnostics-13-02956-f004:**
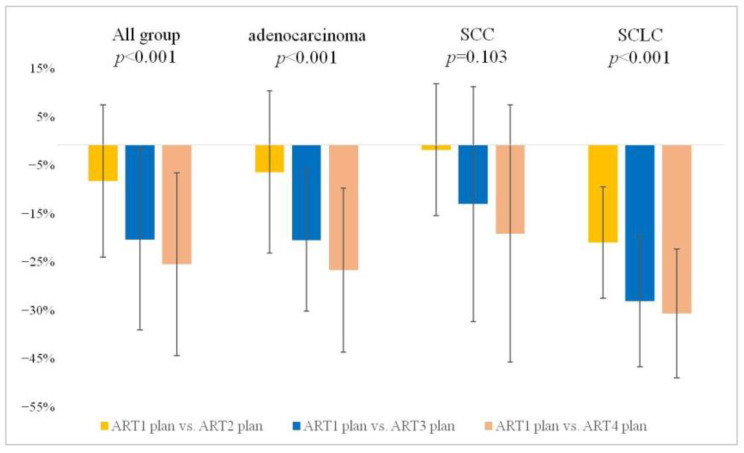
Weekly stepwise tracking of GTV changes from the start of RT. Weekly stage-by-stage gross tumor volume (GTV) changes for all patient groups (All, adenocarcinoma, SCC, and SCLC) are shown. The chart presents the volumetric changes as percentages (%) between the ART1 plan and the ART2, ART3, and ART4 plans based on a cumulative comparison from the start of treatment. This provides insight into how GTV changes evolve over time for each group. SCC, squamous cell carcinoma; SCLC, small-cell lung cancer; ART, adaptive radiotherapy.

**Figure 5 diagnostics-13-02956-f005:**
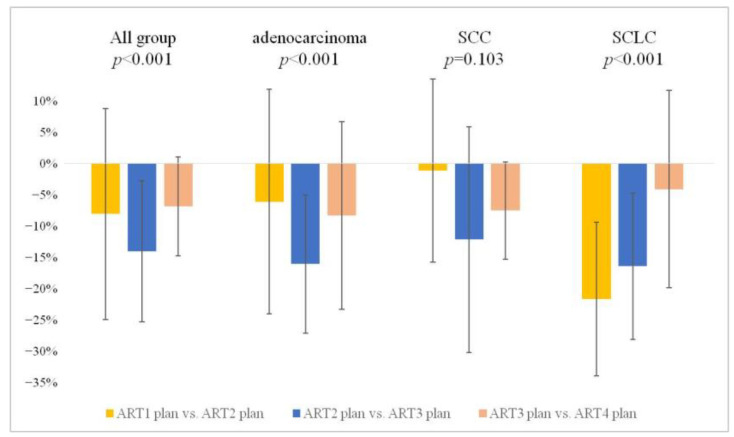
Weekly interval comparison of GTV alterations. The chart illustrates weekly comparative gross tumor volume (GTV) changes for all patient groups (All, adenocarcinoma, SCC, and SCLC). It presents the volume changes in terms of percentage (%) between successive plans: ART1 plan versus ART2 plan, ART2 plan versus ART3 plan, ART3 plan versus ART4 plan, etc. This provides insight into how the GTV changes evolve weekly for each group. SCC, squamous cell carcinoma; SCLC, small-cell lung cancer; ART, adaptive radiotherapy.

**Table 1 diagnostics-13-02956-t001:** Patients’ characteristics.

	Total n = 30	NSCLC n = 20	SCLC n = 10	*p*-Value	Analysis Method
Age	75 (54–89)	71 (54–89)	75 (56–87)	0.227	Student *t*-test
Sex				>0.999	Fisher’s exact test
Male	25 (83.33%)	17 (85.00%)	8 (80.00%)		
Female	5 (16.6%)	3 (15.00%)	2 (20.00%)		
Smoking				0.682	Fisher’s exact test
Yes	22 (73.3%)	14 (70.0%)	8 (80.0%)		
No	8 (26.7%)	6 (30.0%)	2 (20.0%)		
T stage				0.052	Fisher’s exact test
1	9 (30.00%)	4 (20.00%)	5 (50.00%)		
2	8 (26.67%)	5 (25.00%)	3 (30.00%)		
3	9 (30.00%)	9 (45.00%)	0 (0.00%)		
4	4 (13.3%)	2 (10.00%)	2 (20.00%)		
N stage				0.663	Fisher’s exact test
0	14 (46.67%)	9 (45.00%)	5 (50.00%)		
1	4 (13.33%)	2 (10.00%)	2 (20.00%)		
2	10 (33.33%)	8 (40.00%)	2 (20.00%)		
3	2 (6.67%)	1 (5.00%)	1 (5.00%)		
Chemo				0.101	Fisher’s exact test
Yes	20 (66.67%)	11 (55.00%)	9 (90.00%)		
No	10 (33.33%)	9 (45.00%)	1 (10.00%)		
GTV location				0.588	Fisher’s exact test
LUL	8 (26.7%)	7 (35%)	1 (10%)		
LLL	4 (13.3%)	2 (10%)	2 (20%)		
RUL	9 (30%)	6 (30%)	3 (30%)		
RML	2 (6.7%)	1 (5%)	1 (10%)		
RLL	7 (23.3%)	4 (20%)	3 (30%)		
PlanCT to RT interval (days)	3.5 (2–6)	3 (2–6)	4.5 (3–6)	0.177	Student *t*-test

Age and PlanCT to RT interval are presented as mean ± standard deviation, while all other items are presented as numbers (percentages). The *p*-values were calculated using Student’s *t*-test or Fisher’s exact test to test for differences between the NSCLC and SCLC groups. Statistical significance was set at *p* < 0.05. RT, radiotherapy; GTV, gross tumor volume; LUL, left upper lobe; LLL, left lower lobe; RUL, right upper lobe; RML, right middle lobe; RLL, right lower lobe; NSCLC, non-small-cell lung cancer; SCLC, small-cell lung cancer.

**Table 2 diagnostics-13-02956-t002:** Statistical comparison of CI, mDHI, and rDHI differences between the initial plan and ART1, 2, and 3 plans for each patient.

	*p*-Value (All) *	Initial Plan vs. ART1 Plan *p*-Value	Initial Plan vs. ART2 Plan *p*-Value	Initial Plan vs. ART3 Plan *p*-Value	Initial Plan vs. ART4 Plan *p*-Value
CI	0.437	>0.999	>0.999	>0.999	>0.999
mDHI	0.191	0.655	0.123	>0.999	0.554
rDHI	0.211	>0.999	>0.999	0.320	0.686

* Repeated-measures ANOVA was conducted, and the reported *p*-values (all) represent the trend value across the four repeated measurements. CI, conformality index; mDHI, moderate dose homogeneity index; rDHI, radical dose homogeneity index; ART, adaptive radiotherapy.

**Table 3 diagnostics-13-02956-t003:** Comparison of absolute and relative changes in GTV from rCT1 to rCT4 across lung cancer subtypes.

	All Group	Adenocarcinoma Group	SCC Group	SCLC Group
rCT1 GTV abs. (cm^3^)	56.68 ± 66.65	34.21 ± 32.46	86.80 ± 96.91	49.03 ± 47.29
rCT2 GTV rel. (%)	−7.77 ± 16.31	−5.86 ± 17.39	−1.07 ± 14.14	−20.95 ± 11.88
rCT3 GTV rel. (%)	−20.26 ± 19.34	−20.47 ± 15.11	−12.68 ± 25.15	−33.51 ± 14.05
rCT4 GTV rel. (%)	−25.54 ± 19.56	−26.86 ± 17.53	−19.03 ± 27.56	−36.14 ± 13.77
*p*-value All (abs.)	<0.001	0.034	0.023	0.034
*p*-value All (rel.)	<0.001	<0.001	0.103	<0.001
*p*-value rCT1 vs. rCT2 (rel.)	0.010	>0.999	>0.999	0.001
*p*-value rCT1 vs. rCT3 (rel.)	<0.001	0.015	>0.999	<0.001
*p*-value rCT1 vs. rCT4 (rel.)	<0.001	0.009	0.758	<0.001
*p*-value rCT2 vs. rCT3 (rel.)	<0.001	0.031	0.250	0.076
*p*-value rCT3 vs. rCT4 (rel.)	0.134	>0.999	>0.999	>0.999

This table provides an overview of the changes in GTV for each patient group from rCT1 to rCT4. The rCT1 GTV is expressed as an absolute value (cm^3^), whereas the GTVs for rCT2, rCT3, and rCT4 are shown as percentage changes relative to rCT1. The *p*-value All (abs.) signifies the statistical significance of the absolute change in GTV across all rCT stages, while *p*-value All (rel.) indicates the statistical significance of the relative change in GTV across all rCT stages. Subsequent *p*-values represent the statistical significance of the rate of change between each rCT stage and rCT1. The final two rows detail the statistical significance of the rate of change between rCT2 and rCT3 and between rCT3 and rCT4. SCC, squamous cell carcinoma; SCLC, small-cell lung cancer; rCT, resampled computed tomography images.

**Table 4 diagnostics-13-02956-t004:** Comparative analysis of whole lung DVH parameters across ART plans over time.

	ART1 Plan	ART2 Plan	ART3 Plan	ART4 Plan	*p*-Value (All)
ALL Group					
V20_Gy_ (cm^3^)	340.85 ± 158.75	326.22 ± 145.19	309.14 ± 150.94	301.96± 135.19	0.002
V30_Gy_ (cm^3^)	180.20 ± 87.77	169.20 ± 78.05	161.17 ± 75.50	155.45 ± 72.34	<0.001
D_mean_ (Gy)	8.93 ± 3.84	8.41 ± 3.23	8.05 ± 3.26	8.00 ± 3.19	0.004
Adenocarcinoma					
V20_Gy_ (cm^3^)	301.05 ± 156.68	292.07 ± 164.91	273.77 ± 183.51	272.25 ± 149.34	0.188
V30_Gy_ (cm^3^)	162.11 ± 67.00	159.14 ± 71.77	153.95 ± 70.48	145.47 ± 65.90	<0.001
D_mean_ (Gy)	7.97 ± 3.41	7.88 ± 3.12	7.43 ± 3.34	7.43 ± 3.57	0.125
SCC					
V20_Gy_ (cm^3^)	357.44 ± 135.99	345.66 ± 108.04	332.24 ± 119.01	312.24 ± 107.71	0.051
V30_Gy_ (cm^3^)	191.19 ± 85.05	182.67 ± 72.26	171.76 ± 74.17	164.53 ± 70.63	0.071
D_mean_ (Gy)	9.89 ± 3.50	9.28 ± 2.95	8.59 ± 2.61	8.50 ± 2.44	0.202
SCLC					
V20_Gy_ (cm^3^)	364.07 ± 188.39	340.94 ± 164.64	321.42 ± 153.04	321.38 ± 153.20	0.007
V30_Gy_ (cm^3^)	187.31 ± 112.01	165.80 ± 94.49	157.81 ± 87.77	156.34 ± 85.61	0.002
D_mean_ (Gy)	8.95 ± 4.64	8.07 ± 3.71	8.13 ± 3.95	8.08 ± 3.65	0.052

Repeated-measures ANOVA was conducted, and the reported *p*-values (All) represent the trend value across four repeated measurements. DVH, dose–volume histogram; ART, adaptive radiotherapy; SCC, squamous cell carcinoma, SCLC, small cell lung cancer.

**Table 5 diagnostics-13-02956-t005:** Comparative analysis of esophagus DVH parameters across ART plans over time.

	ART1 Plan	ART2 Plan	ART3 Plan	ART4 Plan	*p*-Value (All)
ALL Group					
D_max_ (Gy)	29.16 ± 15.51	29.43 ± 15.57	28.71 ± 15.25	28.59 ± 15.18	0.933
D_mean_ (Gy)	12.57 ± 11.63	13.05 ± 11.35	10.59 ± 9.16	10.78 ± 9.50	0.731
Adenocarcinoma					
D_max_ (Gy)	22.46 ± 13.87	23.28 ± 14.03	23.31 ± 14.67	22.49 ± 14.40	0.391
D_mean_ (Gy)	8.34 ± 8.60	10.71 ± 10.65	8.05 ± 8.86	8.06 ± 8.96	0.197
SCC					
D_max_ (Gy)	29.00 ± 13.50	27.91 ± 13.78	28.02 ± 13.15	28.62 ± 12.71	0.451
D_mean_ (Gy)	12.83 ± 11.20	12.48 ± 9.63	11.96 ± 9.62	12.39 ± 9.81	0.738
SCLC					
D_max_ (Gy)	36.01 ± 17.27	37.11 ± 16.85	34.81 ± 16.93	34.65 ± 17.05	0.687
D_mean_ (Gy)	16.53 ± 14.11	15.97 ± 13.89	11.77 ± 9.42	11.88 ± 10.08	0.828

Repeated-measures ANOVA was conducted, and the reported *p*-value (All) represents the trend value across four repeated measurements. DVH, dose–volume histogram; ART, adaptive radiotherapy; SCC, squamous cell carcinoma, SCLC, small cell lung cancer; D_mean_, mean dose; D_max_, maximum dose.

**Table 6 diagnostics-13-02956-t006:** Comparative analysis of heart DVH parameters across ART plans over time.

	ART1 Plan	ART2 Plan	ART3 Plan	ART4 Plan	*p*-Value (All)
ALL Group					
V20_Gy_ (cm^3^)	40.23 ± 57.88	39.21 ± 55.63	35.91 ± 55.33	37.60 ± 56.79	0.025
V30_Gy_ (cm^3^)	14.84 ± 23.37	14.41 ± 23.62	13.47 ± 22.84	14.08 ± 24.87	0.182
D_mean_ (Gy)	6.31 ± 5.85	5.78 ± 5.35	5.65 ± 5.33	5.56 ± 5.38	0.646
Adenocarcinoma					
V20_Gy_ (cm^3^)	17.58 ± 34.08	20.59 ± 42.71	21.20 ± 43.89	20.13 ± 41.07	0.865
V30_Gy_ (cm^3^)	7.36 ± 16.83	8.80 ± 20.29	9.38 ± 22.14	8.60 ± 19.60	0.171
D_mean_ (Gy)	3.72 ± 4.80	3.27 ± 3.54	3.17 ± 3.73	3.04 ± 4.29	0.166
SCC					
V20_Gy_ (cm^3^)	49.70 ± 67.66	48.49 ± 60.78	44.80 ± 62.21	51.75 ± 72.88	0.457
V30_Gy_ (cm^3^)	16.00 ± 25.42	15.64 ± 26.49	15.29 ± 24.38	18.44 ± 33.36	0.731
D_mean_ (Gy)	7.47 ± 6.02	7.20 ± 6.02	6.69 ± 5.78	6.72 ± 6.02	0.868
SCLC					
V20_Gy_ (cm^3^)	53.43 ± 64.75	48.54 ± 62.26	41.73 ± 60.86	40.92 ± 53.11	0.009
V30_Gy_ (cm^3^)	21.14 ± 26.89	18.80 ± 25.01	15.75 ± 23.84	15.20 ± 20.99	0.003
D_mean_ (Gy)	7.73 ± 6.29	6.88 ± 5.75	7.08 ± 5.82	6.93 ± 5.28	0.644

Repeated-measures ANOVA was conducted, and the reported *p*-value (All) represents the trend value across four repeated measurements. DVH, dose–volume histogram; ART, adaptive radiotherapy; SCC, squamous cell carcinoma, SCLC, small cell lung cancer; D_mean_, mean dose.

## Data Availability

The datasets used and/or analyzed during the current study are available from the corresponding author on reasonable request.

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
