# Peer review of "Evaluating the Necessity of Adaptive RT and the Role of Deformable Image Registration in Lung Cancer with Different Pathologic Classifications"

_diagnostics, 2023, doi:10.3390/diagnostics13182956_

Round 1

Reviewer 1 Report

I would like to thank the Authors for the submitted manuscript.

On average, I appreciate the topic: different timing for ART based on histology in lung cancer.

The results are based on 30 patients, which is a small sample to drive firm conclusions.

The manuscript in results section is verbose, I would suggest making it easier for readers to summarize in a table.

Minor typos:

Abstract: … in radiation dose irradiated to normal tissues… please revise

The study was and approved: typos, delete “and”

Conclusion: underscore, I would suggest “highlight “

no further comments

Reviewer 2 Report

I have read with great interest the manuscript "Evaluating the necessity of adaptive RT and the role of deform-able image registration in lung cancer with different pathologic classification" which concerns an interesting subject. 

I have no major objections, apart from I believe the MS would approve of being shortened for reading purposes. Some of the information in the text could also be presented in tables instead. The limitations are clearly presented. 

Please clarify the nodal status of the patients and if chemotherapy was given or not. 

Tables are missing in the submission. 

It would be interesting to couple the findings to clinical outcome data. 

Good
